# The Preschool Eating, Lifestyle, and Sleeping Attitudes Scale (PRELSA Scale): Construction and Pilot Testing of a Tool to Measure Factors Associated with Childhood Obesity

**DOI:** 10.3390/healthcare11101365

**Published:** 2023-05-09

**Authors:** Jesús Carretero-Bravo, Mercedes Díaz-Rodríguez, Bernardo Carlos Ferriz-Mas, Celia Pérez-Muñoz, Juan Luis González-Caballero

**Affiliations:** 1Department of Biomedicine, Biotechnology and Public Health, University of Cadiz, Avenida Ana de Viya 52, 11009 Cádiz, Spain; 2Department of Nursing and Physiotherapy, University of Cadiz, Avenida Ana de Viya 52, 11009 Cádiz, Spain; 3Clinic Management Unit (UGC), Andalusian Health Service, 11510 Puerto Real, Spain; 4Department of Statistics and Operations Research, University of Cadiz, Polígono Río San Pedro, 11510 Puerto Real, Spain

**Keywords:** childhood obesity, scales, pilot test, parental attitudes, feeding practices, physical activity

## Abstract

(1) Background: Childhood obesity poses a global health challenge. In the period from two to six years, the fundamental risk factors are associated with modifiable habits, related to parental attitudes. In this study, we will analyze the construction and pilot test of the PRELSA Scale, designed to be a comprehensive tool that covers the whole problem of childhood obesity, from which we can later develop a brief instrument. (2) Methods: First, we described the scale construction process. After that, we conducted a pilot test on parents to check the instrument’s comprehensibility, acceptability, and feasibility. We detected items to be modified or eliminated through two criteria: the frequencies of the categories of each item and responses in the Not Understood/Confused category. Finally, we sought expert opinion through a questionnaire to ensure the content validity of the scale. (3) Results: The pilot test on parents detected 20 possible items for modification and other changes in the instrument. The experts’ questionnaire showed good values on the scale’s content, highlighting some feasibility problems. The final version of the scale went from 69 items to 60. (4) Conclusions: Developing scales that detect parental attitudes associated with the onset of childhood obesity may be the basis for future interventions to address this health challenge.

## 1. Introduction

Childhood obesity poses a global health, policy, and research challenge. Obesity among children around the world has risen from just 4% in 1975 to 18% in 2016 [1]. In Europe, 1 in 3 children is currently overweight or obese [2], and it is estimated that by 2030, there will be more than 250 million obese children in the world [3]. It is associated with multiple diseases affecting the whole body and in the long term, obese children are at increased risk of chronic diseases [4,5] as well as being obese adults.

The situation in Spain is similar. More than 50% of the adult Spanish population is overweight and the child population also has a high prevalence [6,7]. Furthermore, it is estimated that in 2030, 18.6% of children between 5 and 9 years of age will be obese [3]. This situation is worrisome since a high percentage of overweight children will be obese in adulthood [8]. Given this reality, it is necessary to develop tools to measure the underlying factors leading to obesity from an early age.

Childhood obesity presents three critical periods in its development: the first thousand days (gestation and two years of life), the adipose rebound (an increase in BMI that occurs at around 6 years of age), and adolescence [9]. Interventions focused on the first period, the period of metabolic programming, have shown that good habits associated with parents can prevent future childhood obesity and, therefore, in adult life [10].

After the first thousand days, the two- to six-year period is when there is no longer as much influence of metabolic programming and prevention should focus on acquiring and maintaining appropriate habits [11]. The importance of this period lies in the fact that it is the time prior to adipose rebound. If the programming in the first thousand days has not been adequate and the children have not acquired good habits afterwards, an early adipose rebound can occur, which implies an increased risk of obesity. The prevalence of childhood obesity stabilizes between the ages of 6 and 9 years, while the period prior to rebound is critical in the appearance of new cases [9].

The risk factors associated with childhood obesity in this period are considered modifiable or educable factors [11]. Childhood obesity has been associated with the type of diet and family behaviors related to food [12], with restrictions on sweet flavors [13], with the time of screen viewing [14], with sedentary habits [15], and with low physical activity [16]. Inadequate sleep patterns and insufficient sleep duration have also been associated with an increased risk of childhood obesity [17,18].

Although the authorities have developed prevention programs, the effects so far have been small, with obesity remaining stable in the period 2011–2019 [7]. Government interventions have focused on adequate child development and growth [19], with general recommendations that do not focus on the specific problems of each child and do not attempt to influence and modify parental attitudes and habits. Research studies have focused on only one or two risk factors for childhood obesity [20,21], which makes obesity prevention incomplete. In addition, they tend to be focused on ages somewhat older than the preschool period and associated with schools, which detracts from the importance of parental thoughts and attitudes in obesity prevention.

Children’s development is clearly influenced by these parental thoughts and attitudes, especially in the case of childhood obesity [22,23]. At this age, parents are one of the main factors that influence the development of behaviors and habits of their children [22,24], not only by their thoughts and attitudes but also by their habits from which the child learns. Some studies have shown that parents directly influence the alleviation of overweight in children by promoting appropriate values and serving as role models [25,26]. A meta-analysis showed that interventions that influence parental behaviors in the preschool stage have good results on BMI [27].

Taking this into account, a good starting point to act on childhood obesity can be to measure parental attitudes and thoughts that may lead to this problem, using precise psychometric instruments associated with the dimensions of interest. The instruments that have attempted to measure these dimensions in Spain have been characterized by being based on specific habits, not so much on attitudes or thoughts [20,21], or being instruments with few items in each dimension [28]. We believe that an instrument of these characteristics should be broad, with a clear objective: to make it as precise as possible initially, to later develop another brief instrument that selects the best items in each dimension. To our knowledge, no reliable and reproducible instruments with these characteristics have been developed in our country to measure all parental attitudes and thoughts on obesogenic factors at preschool age.

These instruments, once validated, can be used by health or educational institutions as a basis for prevention interventions. Therefore, the main objective of this study is to develop an accurate and reliable instrument that analyzes the attitudes and thoughts of parents or caregivers of preschool children on the factors associated with the development of childhood obesity. This article shows the process of construction and pilot testing of the Preschool Eating, Lifestyle, and Sleeping Attitudes Scale (PRELSA Scale), a scale designed to be an extensive instrument that covers the whole problem and from which we can later develop a brief instrument consisting of the most representative items.

## 2. Materials and Methods

This study shows the initial phase of construction of the PRELSA Scale, which is divided into two phases. The first phase includes the design and creation of the questionnaire, and the second phase includes a pilot test of the first version of the instrument. This pilot test was carried out on two samples, a group of parents and a committee of experts with knowledge of childhood obesity.

### 2.1. Phase 1: Design and Creation of the Scale

In the first phase, a research team created an initial version of the scale analyzing existing scales and incorporating items that the team considered necessary in the dimensions to be measured. The team that generated this version consisted of four members: a pediatrician with clinical experience in child obesity (B.C.F-M.), a physician with research experience in child health prevention (M.D-R.), a doctor in statistics with experience in health and scale development (J.L.G-C.), and a researcher in statistics with knowledge in scales (J.C-B.).

To develop an initial conceptual framework for PRELSA Scale, the research team conducted a literature search about the three constructs of interest and the most frequent questionnaires associated with these constructs. Although these are constructs for which there are already validated scales for measurement, we consider developing a new global scale from scratch necessary for two main reasons. First, most scales do not have validation studies in our country, so they need cultural adaptation and translation. Secondly, some novel aspects of the habits of preschool children, such as the influence of screen viewing, had yet to be considered in depth in the existing instruments.

Given the nature of the constructs and their dimensions, this is a reflective framework in which changes in the items and dimensions that define the construct are expected to be due to changes in everyone’s construct [29]. Three meetings were held among the research team, from which we derived the conceptual framework that defines the scale, as shown in Figure 1.

We will now develop in depth the process that led to the conceptual framework in Figure 1 and the origin of the items that will define each dimension. The team tried to ensure enough items to cover all the problematic aspects, intending to initially construct a broad instrument that would cover all the dimensions. As mentioned in the introduction, we will be able to obtain two instruments: an extensive one covering all the problems and a brief one, derived from the extensive one and made up of the most representative items of each dimension.

#### 2.1.1. Feeding Practices and Attitudes

Parental practices associated with feeding have been extensively studied, with questionnaires at different ages of children, from different perspectives (parents, caregivers, and children themselves), and in all types of populations. A review conducted in 2016 [30] defines the dimensions derived from these practices and parental attitudes about feeding their children.

A feeding scale usually consists of three main factors: parental control of feeding, structure in feeding, and autonomy and parental support. In our case, within the preschool age group, the research team decided to change the dimension of autonomy and parental support for another called quality and variety of the diet since the team thinks that the importance given by the parents to a varied diet and quality food is more determinant at this age.

The most widely used and studied instrument is the Child Feeding Questionnaire (CFQ), which has validation studies in Spanish in similar populations [31,32]. Other instruments are the Comprehensive Feeding Practices Questionnaire (CFPQ) [33] and the Feeding Practices and Structure Questionnaire [34], which have initial validation studies in parents of children of similar ages to those of our study.

According to what was seen in these scales, the team decided that in addition to the three dimensions mentioned, a fourth one was added, associated with the parent’s perception of their child’s weight, which also appears in the CFQ, and which has been shown in another study to be a factor of interest in the risk of obesity [35]. Thus, the feeding construct became four dimensions, two of which (Parental Control and Feeding Structure) also consisted of two and three subdimensions, respectively.

#### 2.1.2. Physical Activity and Lifestyle Attitudes

This dimension encompasses, on the one hand, parental attitudes towards children’s exercise and, on the other hand, lifestyles associated with sedentary behaviors and screen viewing. The scales that analyze parental practices in physical activity are varied, although few studies present a validated scale for preschool age. A review conducted in 2020 [36] suggests that an instrument that aims to analyze the construct of physical activity should at least analyze the ability of parents to stimulate physical activity, logistical support for the activity, and co-participation in the activities of parents or caregivers.

Of the tools in the review, the Parenting Strategies for Eating and Activity Scale (PEAS) [37] is of interest to our study because the other two focus on older children and the differences in physical activity between ages are higher than in the other aspects. This scale has also been used in other interventions and validated in other settings [38,39]. Other instruments of interest are the Parent Perceptions of Physical Activity Scale (PPPAS) [40] and the Physical Activity Parenting Practices for Preschoolers—Hong Kong (PAPPP-HK) [41].

Another important aspect associated with physical activity is the environment and the availability of materials and spaces for physical activity in children. We highlight the Home Opportunities for Physical activity check-Up (HOP-Up) instrument [42], which will make it possible to obtain items associated with logistic support for physical activity.

Regarding sedentary habits, screen time is an aspect that the WHO recommendations consider fundamental. The review [43] analyzes the tools that check physical activity and screen time. These tools mainly count screen-viewing frequencies and habits, but there are no validated scales on parent attitudes to our knowledge. We will consider what can be found in [14,44] on parental attitudes about screen viewing.

#### 2.1.3. Sleeping Practices and Attitudes

We found a review [45] of questionnaires that analyze the child’s sleep from a concrete habit perspective. However, on parental practices associated with sleep, there do not seem to be instruments associated with attitudes and practices with extensive validity studies [46]. One of the most widespread instruments is the Children’s Sleep Habits Questionnaire [47], but it mainly analyzes habits and routines, not so many attitudes. Among the instruments associated more with attitudes, we highlight the Sleep Attitudes and Beliefs Scale (SABS) [48] and the Parent–Child Sleep Interactions Scale (PSIS), the latter with some more extensive validation studies [49,50].

From these last two instruments and the contributions of the team, it was established that the sleep construct had two dimensions: the parent’s knowledge of possible sleep modification and the caregiver’s interaction with the child’s sleep.

Considering this review of the most important instruments, the conceptual framework, and the proposed items, the initial version of the PRELSA Scale consisted of 69 items, which were translated into Spanish and adapted in their wording to be associated with parental thoughts and attitudes. We formulated all items according to a Likert response scale from 1 to 5, associated with most items to a disagree/agree scale and some others to never/always. Most of the items were formulated in a positive sense so that the response associated with the appropriate attitude was agreement, although some items (15) were formulated in a negative sense because this was considered the best wording.

#### 2.1.4. Sociodemographic Characteristics and Habit and Behavioral Variables

In addition to the scale and to be able to carry out a complete validation study, the questionnaire consists of three parts. A first section on sociodemographic characteristics of the families, a second section formed by the PRELSA Scale, and finally, a section on specific habits and behaviors of the families associated with the three dimensions.

The first version of the sociodemographic characteristics section consisted of 27 questions in which the family responded to physical characteristics (sex, age, weight, height, school situation, or the number of family members), socioeconomic characteristics (family income, level of studies, work situation, or marital status), characteristics of their environment (square meters of the home, outdoor spaces, or location of the home), and some questions about their relationship with the children of the family. Although this first version contained many sociodemographic questions, the research team considered that they were all related to the dimensions of the scale and that after this pilot test would make it possible to reduce the number of items.

The last section consisted of originally 23 questions in which respondents were asked about specific habits and behaviors of their children. These questions are based on the WHO’s basic recommendations on minimum habits to prevent childhood obesity [51] related to the consumption of sugary foods, hours of physical activity, and hours of screen viewing, among other habits.

### 2.2. Phase 2: Pilot Test

This second phase is a descriptive and cross-sectional study with two objectives: on the one hand, to review the functioning of the scale in the families and, on the other hand, to know the opinion of experts on the scale’s content.

In the first stage, the target population was fathers, mothers, or primary caregivers of children between 2 and 6 years of age. We obtained the sample through convenience sampling among individuals who were users of a health center associated with the research team. Individuals did not receive any payment for taking the survey. Fifty individuals who were contacted started the survey online, of whom we obtained 26 completed responses.

The sample had to complete the three sections of the first version of the questionnaire (v-1) and give their opinion on the items. The aim of this part of the pilot test was to check comprehensibility (ability to understand the items), acceptability (consideration of whether the scale is appropriate to the individual), and feasibility (whether the scale and the questionnaire are possible to complete in their context). For this purpose, each question gave the option of answering if the person did not understand the question. The respondent could also answer at the end of each section if the questions seemed appropriate or if any items could be added or removed. Finally, they could express their general opinion in an open-text format.

Based on the responses and opinions of the parents in this first stage, the research team decided whether to eliminate, transform, or add some items to the scale and the rest of the instrument. As a result, we obtained a new version of the scale (v-2). A committee of experts will revise this new version in the next phase.

The second stage of the pilot test had as its target members of the health system (physicians, nurses, psychiatrists, and psychologists) and university researchers with at least three years of experience in pediatrics and obesity. The sample was obtained by convenience among contacts of the research team and diffusion among health personnel and researchers. Twenty-seven experts completed the pilot test, but we discarded two responses because they reported having less than three years of experience in the field. The final sample size was 25 experts.

After reviewing the content of the scale, the experts completed a questionnaire with a total of 14 items in which they were asked about characteristics associated with the content validity of the scale and the entire instrument, such as its adequacy (appropriateness to the dimensions), the construction of the scale, its feasibility in the context, or possible mistakes in the descriptions or the redundancy information of the questionnaire. In addition, the experts could express their general opinion in an open-text format and the possibility of adding or removing items or detecting items that may be redundant (i.e., items that actually ask about the same thing).

With the experts’ results, the research team met again to derive possible modifications to the scale. After this final analysis, we arrived at the last version of the instrument (v-3), which will be used in a subsequent field test, in which we will review the reliability, dimensionality, and validity of the scale in a larger sample.

### 2.3. Data Analysis

First, we analyzed the results of the pilot test on parents in the v-1 questionnaire. To do this, we calculated the usual statistics (mean and deviations for continuous variables, frequencies, and percentages for categorical variables). We also read the content of the open-text format opinions concerning the scale.

We reviewed the frequencies in the six response categories of the 69 items. The research team established two criteria to review items: (1) they had a percentage higher than 80% in any category (more than 20 responses), or (2) they had two or more responses in the “Not understood/Confused” category. By doing so, we avoid items that do not discriminate well and may not be understood. We reviewed these items to decide on their possible modification or elimination in the final instrument.

In the second part, the experts reviewed the new version of the instrument and completed a short questionnaire with 14 items about content validity. We analyzed the frequencies and percentages of each response category of the content validity questionnaire. We also reviewed the open-text format responses of the experts, detecting possible items to be modified or eliminated and possible items to be added to the instrument to obtain the v-3 version of the questionnaire. We used Microsoft Excel (v2304, Microsoft, Washington, DC, USA) for data management and R (4.2.0, R Foundation for Statistical Computing, Vienna, Austria) for statistical calculations and graphs.

## 3. Results

### 3.1. Pilot Test, Families Phase

The characteristics of the parents in the first phase are given in Appendix A. The mean age is 36.5 years, and the high number of women (88.3%) stands out. More than 90% of the sample say that they are responsible for their children’s nutrition, and all are responsible for their children’s lifestyle habits. In addition, 57.6% of the sample said they had a university education, and 53.8% had more than 2000 euros of monthly income.

We can see in Figure 2 the categorization of the items in the scale based on the type of responses in each subdimension analyzed. Nine items had two or more responses in the Not understood/Confused category (Unclear questions in Figure 2), while 11 items had any category with more than 80% of responses (>80% of responses in single category in Figure 2). Three of the five items presented problems in the Diet Variety dimension. With these results, the research team reviewed 20 items for possible modification or elimination.

We also analyzed the open-text format responses and a short questionnaire on satisfaction with the scale to analyze the feasibility of the v-1 questionnaire (Table 1).

More than 90% of the respondents considered the descriptions acceptable or better, 84.6% considered the format and web acceptable or better, and 57.7% considered overall satisfaction high or very high. These results, together with the analysis of the 20 problematic items, led to a new version of the questionnaire (v-2) with several modifications: changes in the wording of the items, fewer items in the scale (from 69 to 62), and changes in the questions of the other two sections (three questions less in sociodemographic part and one less in the habits questionnaire).

### 3.2. Pilot Test, Experts Phase

The experts reviewed the v-2 questionnaire in the second phase of the pilot test. The sociodemographic characteristics of the 25 experts are given in Appendix A. Most of the sample comprises women (72%) and physicians (more than 50%). The experts had an average of 16.88 years of experience in their field.

Respondents filled out a questionnaire of 14 items associated with the content validity of the v-2 scale to check its appropriateness to the subject matter, the construction of the scale, its feasibility, the possible redundancy of the items, and qualitative information about the scale. The questions and questionnaire results are shown in Figure 3.

The overall score for the adequacy of the scale was excellent for 72% of the experts. Within this section, the item on adaptation and standardization to the target group stood out negatively. Regarding construction, 76% of the experts considered it excellent, although 8% considered the instructions for completing the scale could be improved.

Regarding feasibility, 36% of the experts considered it excellent, while 60% considered it good. The results for feasibility were mainly influenced by the item associated with the completion time of the instrument, which several experts considered to be one of the weak points. Finally, 24% of the experts considered the information on redundancy of the scale (i.e., descriptions of possible similarities between different sections of the questionnaire) excellent, while 72% considered it good.

In addition to these results, the experts provided relevant information about the instrument in open-text format responses (Appendix A). Their opinions, together with the results of the previous questionnaire, made it possible to carry out the final modifications of the scale. Three items were eliminated as redundant, and one new item was added in the activity stimulation section at the suggestion of two experts.

The final distribution of the items after both phases of the pilot test and the scales of origin of the items is shown in Figure 4. The summary of the changes during the piloting is given in Table 2, and the final complete version of the questionnaire (v-3) is given in the Appendix A.

## 4. Discussion

In this study, we have carried out the first pilot phase of a scale associated with the main obesogenic habits (PRELSA Scale). Its main objective is to measure modifiable risk factors associated with childhood obesity from 2 to 6 years of age. As we developed the results, we reviewed the comprehensibility, acceptability, and feasibility in a small group of families, while a group of experts in the field analyzed its content validity.

To our knowledge, this study is a pioneer in developing a psychometric instrument in Spain to cover the main risk factors of childhood obesity from the perspective of parental attitudes. Compared to other studies focused on only one or two risk factors, this scale covers all obesogenic habits, which is fundamental in a multifactorial problem such as obesity [11]. Other studies have shown that interventions focused on various aspects associated with obesity are more effective than those focused on only one [52]. This scale could be the basis for future interventions that broadly detect all the risk factors.

Considering this is a subject on which there are already several scales and instruments on individual dimensions, it was not necessary to develop the scale from scratch. Given that the research team also included health experts in the field, we based the construction of the scale on the search for items already constructed and new contributions in specific dimensions. An essential factor that gives relevance to this scale is that several of the instruments on which it is based do not have extended validation processes in Spain. In addition, the other instruments located have had few items or been based on behaviors and habits, not attitudes [21,28,29]. Our scale, which is more extensive in its conception, can be used to have a broad instrument for situations where time is not an issue and also to obtain a brief instrument for settings such as primary care or schools.

The first version v-1 derived by the research team underwent a first pilot test for parents and experts. In the first phase of the pilot test (parents), we obtained 26 complete responses, achieving a complete response rate of 52%. While this is a good response rate, the convenience contact in this sampling implies that in the future field test, the response rate will be lower, so the modifications derived from this pilot test should be aimed at improving the instrument itself and maintaining this percentage of completed responses, given that the field test will require a much larger sample size.

Considering the characteristics of the individuals, the high number of women (88.5% of the sample) was noteworthy. In general, the response rate in surveys is higher in women [53] and more so in this one, whose main topic is children. This is a complex issue to solve since, when it comes to talking about children’s lifestyles, it is still the mothers who carry the main burden in families [54]. The mean age of the respondents seems logical (36.2 years), reflecting the mean age of motherhood in Spain, 32.4 years [55]. The sample in the pilot test reflects childcare well, with more than 90% claiming responsibility for family care.

The level of education and income of the parents surveyed in this first phase are higher than the average level in the province of Cadiz. Given the survey’s subject matter, associated with habits and lifestyles, people with higher education or income are expected to complete the survey [56]. Given that it is an instrument that seeks to detect inadequate attitudes and considering that the prevalence of childhood obesity is higher in the low-income population [1], we must take measures to achieve a sample as representative as possible of all populations. In the event of being able to access a large number of parents, we could establish quotas to improve the sample’s representativeness.

This first part of the pilot test aimed to analyze the comprehensibility, acceptability, and feasibility of the PRELSA Scale. For this purpose, we analyzed the response frequencies of the 69 items of the instrument on a Likert scale from 1 to 5. Considering the responses, the research team found 20 items to review. Seven were eliminated, four because they were considered too obvious and three because their wording was too confusing. Another 10 items were reworded, three of them for dietary variety, since the original wording was biased (for example, the original DV5 item “I consider it necessary for my child to eat enough” changed to “I consider it necessary for my child to eat enough, even if sometimes it is not a varied diet”). Finally, we kept the original wording of the other three items, so that the scale went from 69 to 62 items.

Regarding the rest of the instrument, several people showed some drawbacks of the web medium used in the open-text questions. In addition, some users considered that the explanations given during the survey could have been more precise. For this reason, we improved the format according to the possibilities of the web medium and we wrote the explanations in a more precise and simple way. We also simplified the section on sociodemographic characteristics, eliminating three questions considered unnecessary.

Overall satisfaction with the survey in this first phase of the pilot test was high or very high in 57.7% of the cases. Although more than 40% showed medium or lower satisfaction, we believe that the changes introduced may improve satisfaction with the survey and, with it, adherence to it. With the changes introduced, we arrived at a version of the instrument prepared for the expert phase of the pilot test, the v-2 questionnaire.

A sample of 25 experts evaluated the content validity of the v-2 instrument. In general, the experts’ experience in the field was high, more than 16 years on average, and we obtained a heterogeneous sample of researchers in the field, being the main group physicians, a fact that we believe is good for ensuring the validity of the scale.

The overall results of the experts’ questionnaire were good. Some sections had somewhat lower results, mainly on the complexity of the items, the structure of the questionnaire, the way of writing the items and their responses, and, above all, the time to complete the instrument. All these are associated with the construction and mainly with the feasibility of the questionnaire, which is the section with the lowest score.

To improve the construction of the survey, and given some of the experts’ comments, the wording of several of the items was modified so that they could all be answered on a Likert scale from 1 to 5 associated with the degree of agreement or disagreement. In addition, the fact that the response format will always be the same will reduce the complexity of the questionnaire [57], which is one of the aspects with the somewhat lower score according to the experts.

These improvements will also influence the feasibility of the questionnaire. However, they do not affect one of the problematic points regarding feasibility mentioned by both experts and parents: the time to complete the survey. Parents in the pilot test took more than 15 min to complete the survey, which could be excessive [58] and cause some parents’ responses to be missed. In addition, the information provided by the respondent may be conditioned by the length of the instrument [59] since the response time to the items decreases as the survey progresses. In this sense, most of the changes that have been made mean less time to complete the instrument (fewer items in the scale, fewer questions in the characteristics section, or standardization of the responses). However, it is still a somewhat long instrument.

As was mentioned during the construction of the scale, the fact that the instrument was broad had a double objective: to cover all the dimensions extensively and to be able to derive from it a brief and precise instrument. For this reason, the research team felt it necessary to sacrifice the response burden (i.e., the effort required to answer the survey) of the initial instrument, bearing in mind that in situations where a long instrument is not feasible, we can use the reduced version obtained. Furthermore, for a short version of a scale to be most reliable and accurate, it must be derived from an extensive collection of items in the field [60].

Finally, although good, the results on possible information redundancy mistakes were lower than other items. This fact prompted the research team to modify the wording to clarify the information that parents receive about potential redundancies in the instrument. In addition, the team considered, after the experts’ opinion, three items redundant, so they were eliminated (two items about unhealthy foods in two dimensions of the feeding section, and one item in the feeding section about screen viewing during meals with a similar item in the sedentary habits section). However, one item of interest was added in the Activity Stimulation section, proposed by one of the experts (“I consider my child’s physical activity to be as important to his or her development as his or her studies”). Thus, the final version of the PRELSA Scale (v-3) that will go to the field test will consist of 60 items.

### Strengths and Limitations

The present study has several strengths. To our knowledge, it is one of the first to create a scale that detects all the possible risk factors associated with childhood obesity in Spain from 2 to 6 years of age. In addition, the piloting has allowed us to cover its weaknesses and have an instrument showing promising characteristics.

Nevertheless, we should bear in mind some limitations inherent to the study. The sample of parents in the pilot test has shown some shortcomings since it had many women, and the socioeconomic status was not representative of the area where the study was carried out. This makes it necessary in the future field test to take strategies that allow avoiding, as far as possible, these biases.

In addition, one of the limitations is the inconvenience generated by the long time required to complete the survey. Although, as we have mentioned, this is a fact that the research team has considered necessary, we should take this aspect into account to limit the possible loss of responses that will be caused in the final sample.

## 5. Conclusions and Future Work

Obesity is a multifactorial health problem, and its prevention is necessary from the earliest stages of development. The instruments, such as scales, that detect parental thoughts and attitudes associated with the onset of childhood obesity can provide a good basis for developing future interventions. Although for this to happen, the instruments to be developed need to be created following a precise and accurate methodology.

The PRELSA Scale has followed a piloting process that has allowed the detection of possible flaws and weaknesses in its conception and has shown the validity of its content in this first stage. In the field test on a large sample of parents, we will review the reliability, dimensionality, and validity of the scale. Once these psychometric properties have been confirmed, an analysis will be carried out through the Item Response Theory, which will be the basis for constructing the reduced version of the questionnaire, with a clear final objective: to obtain two validated versions of the instrument, one complete and the other reduced, but retaining the items that best represent the dimensions of interest.

These two instruments can be used in the future to detect where interventions in childhood obesity can be oriented so that they are as effective as possible, with the main focus on families as a factor of change and prevention in this epidemic of the 21st century.

## Figures and Tables

**Figure 1 healthcare-11-01365-f001:**
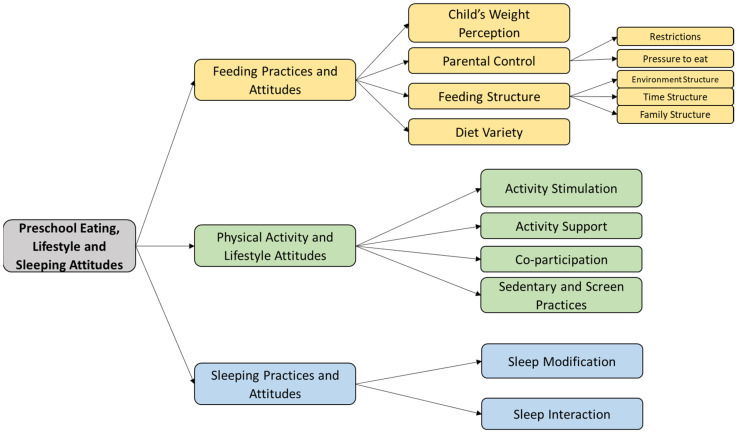
Conceptual framework of constructs and dimensions in the PRELSA Scale.

**Figure 2 healthcare-11-01365-f002:**
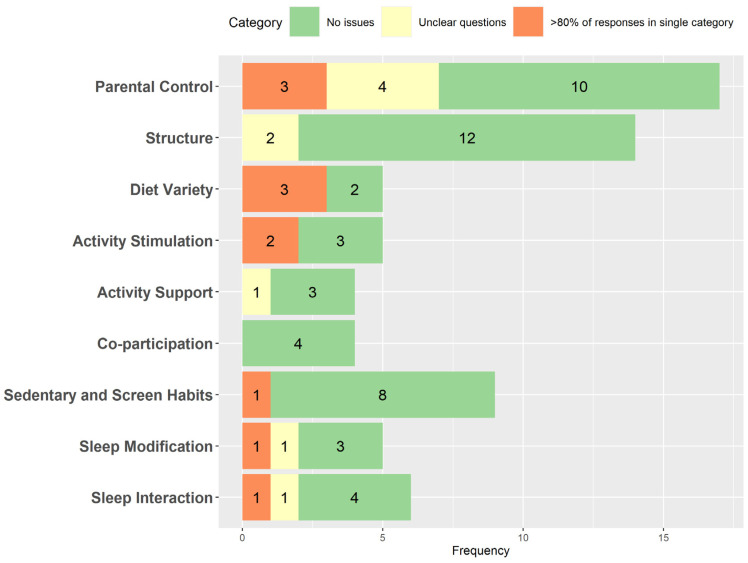
Categorization of items by type of response in the pilot test.

**Figure 3 healthcare-11-01365-f003:**
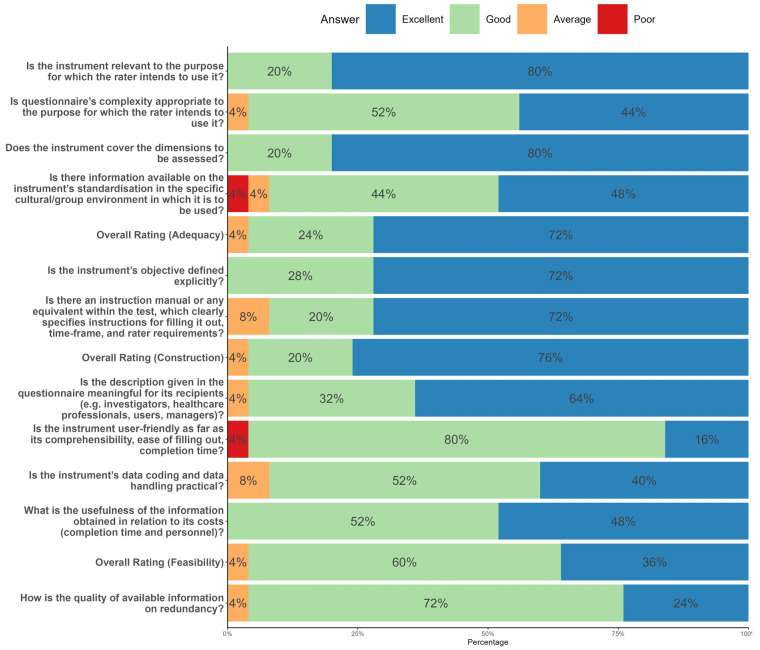
Results of the experts’ questionnaire.

**Figure 4 healthcare-11-01365-f004:**
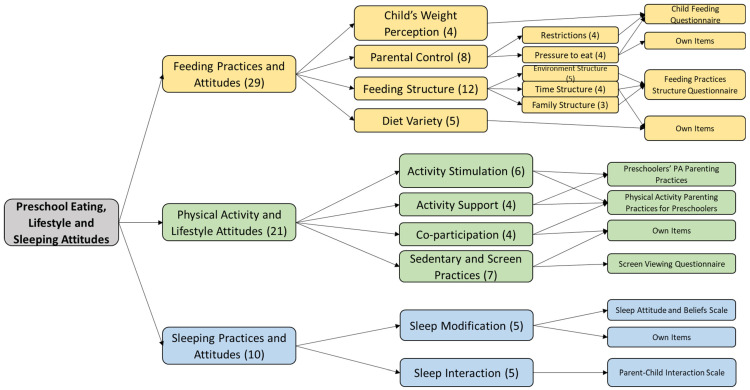
Final distribution of items in the conceptual framework of the PRELSA Scale.

**Table 1 healthcare-11-01365-t001:** General satisfaction with the PRELSA Scale.

Question	Answer	n	%
Are the descriptions and explanations given during the survey clear and sufficient to understand everything correctly?	Very Poor	1	3.8%
	Poor	1	3.8%
	Average	10	38.5%
	Good	12	46.2%
	Excellent	2	7.7%
How would you rate the format, web media, and structure used in the survey in terms of being user-friendly and allowing you to answer everything in an appropriate manner?	Very Poor	0	0.0%
	Poor	4	15.4%
	Average	11	42.3%
	Good	6	23.1%
	Excellent	5	19.2%
Do you consider that the time used to conduct the survey is appropriate to their context?	No, it is not enough	0	0.0%
	No, it is too much	7	26.9%
	Yes, it is adequate	19	73.1%
Please indicate your overall satisfaction with the survey you have completed.	Very Low	2	7.7%
	Low	0	0.0%
	Moderate	9	34.6%
	High	12	46.2%
	Very High	3	11.5%

**Table 2 healthcare-11-01365-t002:** Changes in the questionnaire during all phases of the pilot test.

Section	V-1 Initial	V-2 After Parents’ Phase	V-3 After Experts’ Phase
Sociodemographic	27	24 (3−)	24 (=)
PRELSA Scale	69	62 (7−)	60 (3−, 1+)
Habits questionnaire	28	27 (1−)	26 (1−)

## Data Availability

Data sharing not applicable due to Data Protection.

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
