# Peer review of "The Preschool Eating, Lifestyle, and Sleeping Attitudes Scale (PRELSA Scale): Construction and Pilot Testing of a Tool to Measure Factors Associated with Childhood Obesity"

_healthcare, 2023, doi:10.3390/healthcare11101365_

Round 1

Reviewer 1 Report

This was an interesting article, explaining initial steps in creating a scale that examines parental attitudes towards factors that influence childhood obesity among children aged 2-6 in Spain; namely, eating, physical activity, and sleep.

Overall comments:

·       I am cognizant of the difficulties writing in a language other than one’s first language, but I would strongly recommend the authors have a native English speaker familiar with the topic edit the paper.  There were several sentences that were very difficult to understand, and in places, I felt the English detracted from the argument.

·       In general, I would suggest being much more specific about what was asked iin the survey and what the responses were.  For example, the authors talk about ‘feasibility’ but never say what they mean by this, other than respondent burden.  What do they mean by feasibility/what did they ask? 

·       It would be helpful if the authors were clearer about where this study fits into subsequent, follow-on studies, and to state this at the start.  What are next steps?  For example, there is no attempt in this study to conduct any reliability or validity analysis.  Will this be done in the next study – with a larger and more representative sample?

·       I would also suggest that the authors be clearer about the value of this scale.  If there are already existing scales on each of the three key constructs, the authors need to present a compelling argument as to why not just use those scales.  And relevant to this – there does not seem to be any analysis re: whether this scale is unidimensional or not (or whether it is expected to be).  Presumably this analysis will be done at the next stage, with a larger sample(?)  But at the moment, its value is not as clear as I think it should be.

Specific comments:

Lines 38-39: Presumably you don’t mean to say that this tool will prevent obesity!  I think you mean that it will allow accurate measurement of underlying factors that contribute to childhood obesity, or something.

Line 42: Please explain the adipose rebound – what it is and when it happens.

Lines 80-81: This sentence is hard to understand

Lines 91-92: I would state this earlier, and in the abstract.

Lines 100-101: Did you also review the relevant literature to derive the key constructs, or just existing scales?  I would think you would want to do both.

Fig. 1 and elsewhere: Please fix typo’s (e.g. “Child’s Weight”)

Fig. 1: “PreSchool Eating and Lifestyle Attitudes” doesn’t include either physical activity (is this “lifestyle”?) or sleep.

Line 116: I suggest letting the reader know earlier that this will be coming.

Lines 131-133: Is this based on evidence?

Line 145: What does “lifestyle” refer to?

Line 193: That’s a lot of demographic questions!  Are all of these necessary, particularly given the concerns re: survey length?  How were the items chosen?  Are they based on prior research showing connections to childhood obesity and/or attitudes of parents?  I think you need to justify their inclusion.

Line 200: Some examples of these would be helpful.

Line 203: You don’t say anything about this study receiving Ethics approval.  This needs to be stated.

Lines 210-211: Why were there so many incompletes?  Did participants receive any payment for their time?

Lines 213-216: Did you consider doing cognitive testing to understand how participants were thinking about and answering each question?

Lines 224+: I would suggest making it clearer that you revised the initial instrument based on feedback from the parents/carers, and then provided the revised instrument to the expert panel.

Lines 235 and elsewhere: The discussion of redundant items is unclear.  Could you please provide an example or two? 

Line 240: I’m wondering why there was no exploration of reliability, validity, or scale dimensionality?  I suspect the reason is the small and unrepresentative sample size, and that this is planned at the next stage.  If so, it is important that you state this.

Lines 245-246: I’m wondering if you considered, or will consider in the next study, Rasch Analysis to examine functioning of response categories?

Lines 254+: In general, I would like to see a lot more specifics/examples of results.

Lines 259-260: How does this compare with this age group as a whole in Spain?

Fig 2.  I didn’t understand this figure.  What does “Normal” mean?  What does “Answers in the same category” mean?  What do the categories and numbers in the bars mean?  Does this mean, for example, that there were 17 items measuring Parental Control, and, of these, for 4 items, at least one participant said the question was unclear?  Is there a way to make this figure clearer?

Line 265: How many is “too many”?

Line 265-266”: Is “Not understood/Confused” the same as “Unclear questions” from the figure?

Line 266: What does “too much frequency” mean?

Table 1, first question: Did participants understand what “descriptions and explanations” meant?

Table 1, second question: This is a multi-barrelled question and therefore difficult to answer.  And what does “format” mean?

Table 1, Answer 3: These are confusing response options.

Table 1, Question 4: Did respondents understand that this was a scale they were responding to?  Why not use the term, “questionnaire”?

Lines 279-280: Suggest letting the reader know this sooner

Line 285: Such as?

Line 295: What did you ask re: feasibility?

Lines 298-299: I didn’t understand this.

Lines 305-306: It would be helpful to have a table indicating changes made at each stage.

Line 310: Much of the Discussion repeats the Results. 

Lines 312-313: An instrument can’t prevent anything.

Lines 319-320: So, why not just administer multiple scales?  The authors need to make a stronger/clearer argument about why this new scale is needed.

Lines 325-326: This needs to be explained much earlier in the paper.  I’m also not sure I understand exactly what the authors mean by a “qualitative part”.

Lines 344-359: Did the authors consider implementing some quotas in order to ensure a more representative sample?

Line 362: I didn’t understand this.

Lines 367-368: Some examples would be helpful here.

Lines 376-380: This may not necessarily improve satisfaction.

Lines 379-381: Repeated from Results.

Line 396: I think you mean “survey” here rather than “questionnaire”.

Line 409: If by “feasibility” you mean response burden, you should probably use this term.  If you mean something else/in addition, it would be helpful to define what you mean by “feasibility” and how it was measured.

Lines 424-425: Would this be expected to be different in future surveys?

Line 434: “Multifactorial” or “multifaceted”?

Line 437: I don’t think this is quite what you mean.  I assume this refers to providing a basis upon which interventions can be developed(?)

Line 439: Any literature to support your approach?

Author Response

Thank you for your review. You can find the full response to your comments in the attached file.

Reviewer 2 Report

Thanks to the authors for their submission to HEALTHCARE. The aim of this manuscript
is to try to determine a tool applicable in Spain for the prevention of childhood obesity
by investigating modifiable risk factors in early childhood related to parental attitudes
and practices. I fully acknowledge the time and effort involved in carrying out the
exhaustive search and screening of the different existing tools and questionnaires on this
topic, analysing and synthesising the results, and subsequently drafting the manuscript.
The manuscript meets the vast majority of the required criteria and its writing is clear and
concise despite the complexity and breadth of the procedure used. Even so, some minor
changes should be addressed by the authors.
Kind regards,

Author Response

(The authors gave the same response as above.)

Reviewer 3 Report

First of all, I have to say that I found this manuscript interesting enough to be considered for publication in this journal. However, before doing so, the authors should take into account a series of considerations:

Format:

-       The manuscript does not follow the Healthcare journal format. An erroneous template referring to another publisher's journal is used.

Abstract:

-       The statistical method used is not mentioned.

Keywords:

-       The word "prevention" is repeated in the title of the article. It is recommended to remove and include some other keyword that may help the search for this article once it is published. A good option would be to include "physical inactivity" as it is considered a clear risk factor for obesity.

Introduction:

-       Given the influence of children's and adolescents' level of physical literacy on whether they are overweight or obese, the authors of this article are encouraged to learn more about this novel construct. It would be interesting if it could be introduced in the introduction as well as in the scale itself or at least in future lines of research.

Discussion:

-       It is necessary to include the practical implications of the study and the future line research.

Author Response

(The authors gave the same response as above.)

Reviewer 4 Report

This is a methodological study, which was designed very carefully and appropriately. A more representative sample of the different groups in the population would have been more appropriate. However, in the discussion the authors discussed its implications.

It is a pity that the authors didn't pay enough attention to small details.

1.       I had checked few references which I had doubts about their relevance, and I realized that they were not relevant to where it was quoted. Please check references: 10, 11 and 54. I suggest checking all other references as well in order to be sure that their content is appropriate to the context.

2.       Figure 1 presents the conceptual framework of the construct and dimensions, but, in my opinion, it is misleading. All the arrows should have been in the opposite direction. As the figure represents the components from which each dimension is composed of, it should not present that direction of the association. I suggest leaving the line between the components without an arrow. The same comment is relevant for figure 4.

3.       Figure 2 does not have any reference in the text, so it is not so clear what is its added value.

4.       The results of the expert phase are described in figure 3, however, in the text (row 288) it refers to figure 2.

5.       It seems as the authors didn't proofread their article before submitting it. One example is mentioned above (comments 3,4) another example is present on rows 337-338.

Author Response

(The authors gave the same response as above.)

Round 2

Reviewer 1 Report

I appreciate the authors addressing my comments and revising the paper accordingly.  The paper is much clearer (to me) now.

There are a few remaining issues that I recommend the authors consider.

1.      The authors have added the following sentence to the abstract: “With the pilot test results on parents and experts, we detected items to be modified  or eliminated by analyzing the frequency of category responses.”  Yet, in lines 267-269, they say: “We reviewed the frequencies in the six response categories of the 69 items. The research team reviewed items with two conditions: 1) they had a percentage higher than 268 80% in any category, or 2) they had two or more responses in the "Not understood/Confused" category.”  In other words, in the abstract they state there was just one criterion on which they determined whether to modify an item, and later in the paper they state there were two.  Which is true? (I suspect it’s the latter)

2.       Why did the authors choose the threshold of 80% of responses in a single category for the decision to delete or modify an item?  Is there a reference for this decision or was it just arbitrary?  I think it at least warrants an explanation.

3.       The explanation for Figure 2 is much clearer.  However, the figure is not.  I suggest labeling the green box as “no issues” rather than “normal”; and the red box as something like “>80% of responses in single category” rather than “Answers in the same category” (assuming that this is what this means; I wasn’t sure).

4.       Line 323: I don’t understand what “information on redundancy” means.  Do you mean their evaluation regarding the amount of redundancy across items in the survey?

5.       Finally, and I realise I’m being a bit pedantic, but can you please look up the definitions of “questionnaire” and “survey”.  They are not synonyms.  They are used incorrectly in several places in the manuscript (i.e., “survey” is used when the correct word is “questionnaire”, and visa-versa).

Author Response

Thank you for your review. You can find the answer to it in the attached file.
